# Exploring Neural Scaling Law and Data Pruning Methods For Node Classification on Large-scale Graphs

## ABSTRACT

Recently, how the model performance scales with the training sample size has been extensively studied for large models on vision and language related domains. Nevertheless, the ubiquitous node classification tasks on web-scale graphs were ignored, where the traits of these tasks, such as non-IIDness, semi-supervised setting, and distribution shift, are likely to cause different scaling laws and motivate novel techniques to beat the law. Therefore, we first explore the neural scaling law for node classification tasks on three large-scale OGB datasets. Then, we benchmark several state-of-the-art data pruning methods on these tasks, not only validating the possibility of exploiting data redundancy for improving the original unsatisfactory power law but also gaining valuable insights into a hard-and-representative principle on picking an effective subset of training nodes. Moreover, we leverage the semi-supervised setting of node classification to propose a novel data pruning method, which instantiates our principle in a test set-targeted manner. Our method consistently outperforms related methods on all three datasets. Meanwhile, we utilize a PAC-Bayesian framework to analyze our method, extending prior results to account for both hardness and representativeness. In addition to a promising way to ease GNN training on web-scale graphs, our study offers knowledge of the relationship between training nodes and GNN generalization.

## 1 INTRODUCTION

In recent years, more and more neural scaling laws have been observed [10, 11, 15, 29, 39], within which the power law, characterizing how test error falls along with the increasing amount of training data, has gained much attention. It has been theoretically shown that such power law scaling can be beaten by pruning training data with an appropriate fraction of the hardest examples reserved [30]. Meanwhile, several state-of-the-art data pruning methods [4, 5, 24, 25, 28, 32] were shown to be capable of beating the power law scaling on large-scale image classification datasets. Such theoretical and empirical results provide people with a promising path toward reducing the overhead of model training, which is helpful, especially when the training data is on a web scale.

However, most existing scaling law investigations concentrate on data types such as images [13, 30] and text [8, 12]. Yet, despite its prevalence in web applications such as user modeling on social networks [31] and fraud detection on transaction networks [20], node classification on large-scale graphs remains underexplored. In practice, the number of nodes in such graphs can reach a billion or even trillion orders of magnitude. Such a scale often forbids people to train graph neural networks (GNN) via traversing on all training nodes for tens of epochs. If an unsatisfactory scaling law for node classification tasks is observed, which suggests exploiting data redundancy, then pruning less valuable training nodes might be promising for accelerating GNN training.

Directly borrowing the results from classification tasks on other data types might be unreliable. Traits of node classification distinguishing it from classification tasks on other data types include: (1) The training examples, i.e., nodes, are not independent and identically distributed but associated by graph structures [40]; (2) Most often a semi-supervised learning setting is considered, where the testing nodes except for their labels are accessible during the model training stage [18]; and (3) The training and testing sets are often split in a natural way such as by the time each node emerges, resulting in a distribution shift between training and testing nodes [14].

Thus, it is necessary to deliberately explore the neural scaling law and data pruning methods dedicated to node classification tasks. To this end, we attempt to answer the research questions as follows:

(1) Does the relationship between the classification error rate and the number of training nodes also obey a power law? If this is the case, does the exponent in the power law suggest a satisfactory scaling?

(2) If the scaling law is unsatisfactory, will the state-of-the-art data pruning methods beat it as on other data types? Is there any general principle for picking an effective subset of training nodes?

(3) Can we exploit the traits of node classification tasks to design a more effective data pruning method?

At first, we conduct extensive empirical studies concerning this sample complexity-related neural scaling law for node classification on three large-scale datasets of OGB [14]. A power law scaling is consistently witnessed across the datasets with small exponents, which is unsatisfactory and implies redundancy in the training set. Hence, we implement a benchmark suite to evaluate several data pruning methods comprehensively, some of which enable a sample complexity better than these power law scalings. Through carefully analyzing three kinds of statistics of the nodes preferred by each method, we derive the hard-and-representative principle for picking an effective subset of training nodes.

Furthermore, we instantiate our hard-and-representative principle in a test set-targeted way. Specifically, we formulate node selection as a variant of the $k$-center problem and propose an approximation algorithm with a performance guarantee to solve it, where both hardness and representativeness are measured regarding the test nodes. Empirically, our proposed method is compared to related baselines and shows advantages consistently across these three datasets. Theoretically, we exploit a previous PAC-Bayesian framework [23, 26] to analyze the generalization discrepancy for the subsets determined by our method and discuss the regime, wherein data pruning as a regularizer tends to outperform the entire dataset.

By answering these questions, our studies not only contribute to reducing the computational burden of GNN training on web-scale graph but also provide insights into the connection between training nodes and GNN performance, which tends to help people understand what and how a GNN learns.

## 2 BACKGROUND AND RELATED WORKS

**Node classification**. A node classification task is often considered in a semi-supervised setting [18], where a graph $\mathcal{G} = (\mathcal{V}, \mathcal{E})$ is given, with all its node features $X_{|\mathcal{V}| \times f}$ and the labels of a portion of nodes accessible during the model training stage. We call these labeled nodes training nodes and denote them by $\mathcal{V}^{(\text{tr})}$ while regarding those without labels as test nodes and denoting them by $\mathcal{V}^{(\text{ts})}$. Then, we use $n$ and $m$ to indicate the cardinality of $\mathcal{V}^{(\text{tr})}$ and $\mathcal{V}^{(\text{ts})}$, respectively. For simplicity, we index $\mathcal{V}$ so that the first $n$ nodes belong to $\mathcal{V}^{(\text{tr})}$, and the remaining $m$ nodes belong to $\mathcal{V}^{(\text{ts})}$. Conventionally, we use $A$ to denote the adjacency matrix, where $A_{i,j} = 1$, if $(i, j) \in \mathcal{E}$, otherwise $A_{i,j} = 0$; and we use $D$ to denote the degree matrix, where $D_i = \sum_j A_{ij}$. Their counterparts, wherein one self-loop has been added to each node, are denoted by $\tilde{A}$ and $\tilde{D}$, respectively.

Generally, a GNN predicts node labels based on both node features and graph structures: $\hat{Y}_{|\mathcal{V}| \times |\mathcal{Y}|} = h(X, A; \theta)$, where $\mathcal{Y}$ denotes the label set. At the core of GNN is the message-passing paradigm, where, in each layer, messages are calculated by transforming current node representations, and node representations are updated by aggregating incoming messages from each node's neighbors. Noticeably, a genre of GNNs decouples feature transformation from message propagation [3, 6], among which SGC [34] is highlighted by its popularity in the literature of GNN's theoretical analysis [16, 38]. In this paper, our analysis also focuses on the case of SGC. The learnable parameters of the GNN (i.e., $\theta$) are often optimized by minimizing a loss function of $\hat{Y}_i$ and $y_i$, $i = 1, \ldots, n$, such as a margin loss $l^\gamma(Y_i, y_i) := \mathbf{1}_{\hat{Y}_{i, y_i} \leq \gamma + \max_{j \neq y_i} \hat{Y}_{i,j}}$, where $y_i \in \{1, \ldots, |\mathcal{Y}|\}$ is the $i$-th node's label, $\gamma \geq 0$ denotes a specified margin, and $\mathbf{1}_{[\cdot]}$ represents the indicator function.

**Neural Scaling Laws**. Recently, particularly since the transformer-based large models have become de facto solutions in many domains, more and more observations reveal that test loss often decreases along with the number of model parameters, the amount of computation, and the number of training examples following a power law [8, 10–13, 15, 29, 39]. Beyond phenomenon, [30] theoretically show that, in a teacher-student perceptron setting, the test performance such as error rate $r$ drops as the number of training examples $n$ increases, following a power law $r = n^{-v}$, and a faster drop can be achieved by reserving an appropriate ratio of the hardest training examples. As their setting differs from node classification tasks, and their empirical studies are all conducted on data types other than graph, we deliberately explore the neural scaling law for node classification tasks in this paper.

**Data Pruning**. Since the exponent $v$ in $r = n^{-v}$ is usually close to zero, indicating redundancy in the data, pruning becomes a promising strategy for reducing the overhead of model training. EL2N [28] uses the Euclidean distance between predicted probability distribution and the one-hot label to reflect the magnitude of the gradient norm, where the larger, the harder an example is. Memorization [5] estimates the improvement in the predicted probability of a training example's truth class if this training example is included in the training set, which can be interpreted as the hardness of correctly predicting this example based on the rest of this training sample. Its variant, Influence [5], calculates the memorization score regarding how a training example affects a test example. DDD [24] uses the

number of incorrect predictions made by a pool of models to reflect the hardness of each training example. Forgetting [32] counts the number of mis-predicting the learned label during a training course to reflect the hardness of a training example, where the larger, the harder it is. Active [4] uses the uncertainty in an ensemble model's prediction to reflect the hardness.

Those methods mentioned above are designed for general data pruning purposes. As for node classification tasks on graphs, some active learning methods have taken the traits of tasks and the message propagation nature of GNN into consideration [1, 36, 41]. However, instead of iterations of incremental training and selection, we explore data pruning in this paper, that is to say, picking a subset of nodes from the huge and possibly redundant training set in a single shot. Thus, the purpose is not to save efforts of labeling. Instead, we seek a subset that can lead to an error rate as low as an entire training set.

## 3 OBSERVATION OF A POWER LAW

We conduct our investigation on three node classification datasets of OGB [14], namely, ogbn-products, ogbn-papers100M, and MAG240M. They correspond to large-scale graphs with around 2.4M, 111M, and 121M nodes. More statistics of these three datasets are provided in Appendix (see Table 1). For the generality of our observation, we apply GraphSage [9] to ogbn-products, SGC [34] to ogbn-papers100M, and GAT [33] to MAG240M, where OGB's official implementations are adopted. More details about the applied GNN models can be found in Appendix A.

To observe how the performance of a learned model changes across different numbers of training nodes, we randomly select a fraction of training nodes, where the considered fraction ranges from 20% to 100%, with increments of 10%. Meanwhile, all provided test nodes are used for performance evaluation.

We present the results in Figure 1, where we plot the performances of random pruning at considered pruning rates (denoted by "Random") and a fitted power function (denoted by "Fitted"). As can be seen, these two curves exhibit remarkable similarity, suggesting that the performance scales with the number of training nodes following a power law. Thus, we provide a positive answer to the first research question raised in Section 1, which can be analytically expressed as $r = n^{-v}$, where $r$ denotes the error rate on test nodes, $n$ denotes the number of training nodes, and $v$ characterizes the speed of change.

On all these three datasets, the scaling is unsatisfactory due to the near-zero exponent $v$. On all these three datasets, the scaling is unsatisfactory due to the near-zero exponent $v$. Slow scaling implies that a significant fraction of training examples and computations are allocated for marginal performance gains. For the web-scale graphs in realistic scenarios, avoiding such diminishing returns is appealing, namely, eliminating the necessity for traversing this fraction of data so that lots of computation resources are saved.

## 4 DATA PRUNING FOR NODE CLASSIFICATION TASKS

### 4.1 Benchmarking Related Methods

A promising strategy for improving the scaling is to exploit the redundancy in large-scale graphs, expecting to train a GNN from

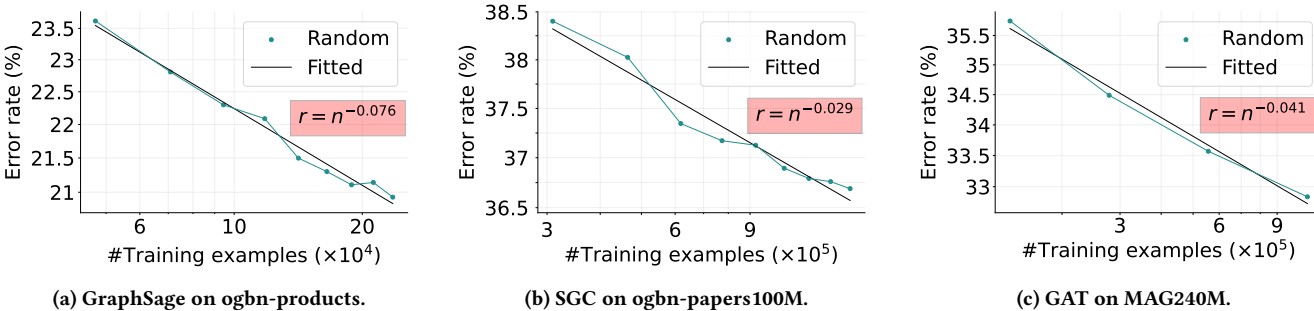

(a) GraphSage on ogbn-products.          (b) SGC on ogbn-papers100M.          (c) GAT on MAG240M.

**Figure 1: Error rate ($r$) v.s. #training nodes ($n$): both axes are in logarithm scale, and thus these near linear relations imply a power law scaling.**

just a portion of labeled nodes such that the learned GNN possesses comparable performance as that learned from all training nodes. Data pruning methods are designed for such a purpose, and some active learning methods can be adapted to this one-shot pruning setting. As prior works have not comprehensively investigated related methods on large-scale node classification tasks, we decided to benchmark several representative methods on ogbn-products, ogbn-papers100M, and MAG240M.

We largely follow the setting adopted in [30]. Specifically, we evaluate general data pruning methods including EL2N [28], Memorization (Mem) [5], Influence score (Infl-max) [5], and DDD [24] as well as graph-dedicated active learning method AGE [1]. Here, we also evaluate our proposed method, which will be detailed in Section 5. It is worth noticing that not all adopted methods are practical. Mem needs training almost a thousand GNN models, which introduces computational overhead larger than the original learning task on the full dataset. Infl-max needs to know the labels of test nodes, which are obviously unavailable in realistic scenarios. We include all these methods because our purposes of benchmarking them are not only to seek the most effective one for node classification but also to understand intrinsic connections between training nodes and model performance.

As each adopted method is designed to assign a score to each training example to reveal its usefulness, we implement each method and let it process the data in a unified way, i.e., to produce a ranking list of training nodes in descending order of their assigned scores. To fairly compare considered methods, we build a pipeline that takes a specified portion of the top-ranked training nodes from a specified ranking list, learn a GNN model from these selected training nodes, and evaluate the learned GNN.

All our code for benchmarking is available at here. The ranking lists generated by considered methods are also included in our repository so that people no longer need to train thousands of GNNs on those large-scale graphs for re-generating the ranking lists. More implementation details about our benchmark suite are deferred to Appendix A.

## 4.2 Results and Findings

We present experimental results in Figure 2. Overall, whatever the dataset and model architecture are, there always exists data pruning methods whose corresponding curve is below that of random

pruning. Thus, we can answer the second research question raised in Section 1: it is always possible to beat the power law scaling by data pruning, where some methods can consistently achieve a better sample complexity while some others may not.

As not all compared methods can successfully beat random pruning, a series of questions naturally arise: are there commonalities amongst successful ranking lists, what might be the reason for some ranking lists' failure, and can we attain some general principle for data pruning in node classification tasks? To gain insights into these questions, we compute the mean degree, mean unnormalized PageRank score (denoted by "Pr") [27], and mean homophilic level (denoted by "Homo") [3] for each specific group of training nodes, where groups corresponding to different quantiles in a ranking are considered. Conventionally, the homophilic level of node $i \in \mathcal{V}$ is defined as $\frac{|\{j|A_{i,j}=1 \wedge y_i=y_j\}|}{|\{j|A_{i,j}=1\}|}$.

As expected, none of these statistics monotonically changes along with quantile since no method ranks training nodes directly and merely by one of these node properties. Therefore, we focus on the groups corresponding to the top 20% of nodes ranked by adopted methods, respectively, and their statistics are illustrated in Figure 2d. Examining the commonalities and distinctions among different methods, we present our findings as follows:

**Successful pruning prefers hard examples.** The mean homophilic level of all training nodes is 0.808, 0.976, and 0.502 on these three datasets, respectively. Methods successfully outperforming random pruning always pick their top 20% of nodes so that the mean homophilic level is remarkably below the average level over the whole graph. On ogbn-products, all adopted methods outperform random pruning and correspond to a lower homophilic level. On ogbn-papers100M, successful ones, including Mem, Infl-max, and Ours have 0.168, 0.255, and 0.191 as their mean homophilic level, respectively. On MAG240M, all adopted methods except for EL2N and DDD have beaten random pruning, and all correspond to a lower homophilic level.

It is generally believed that common GNNs, such as the adopted GraphSage and SGC, benefit from homophily and struggle with predicting nodes with heterophily [22, 37, 42, 43]. Hence, this commonality of successful methods, i.e., a preference for nodes with lower homophilic level, validate the effectiveness of choosing hard training examples. Regardless, we are not saying that hardness is a

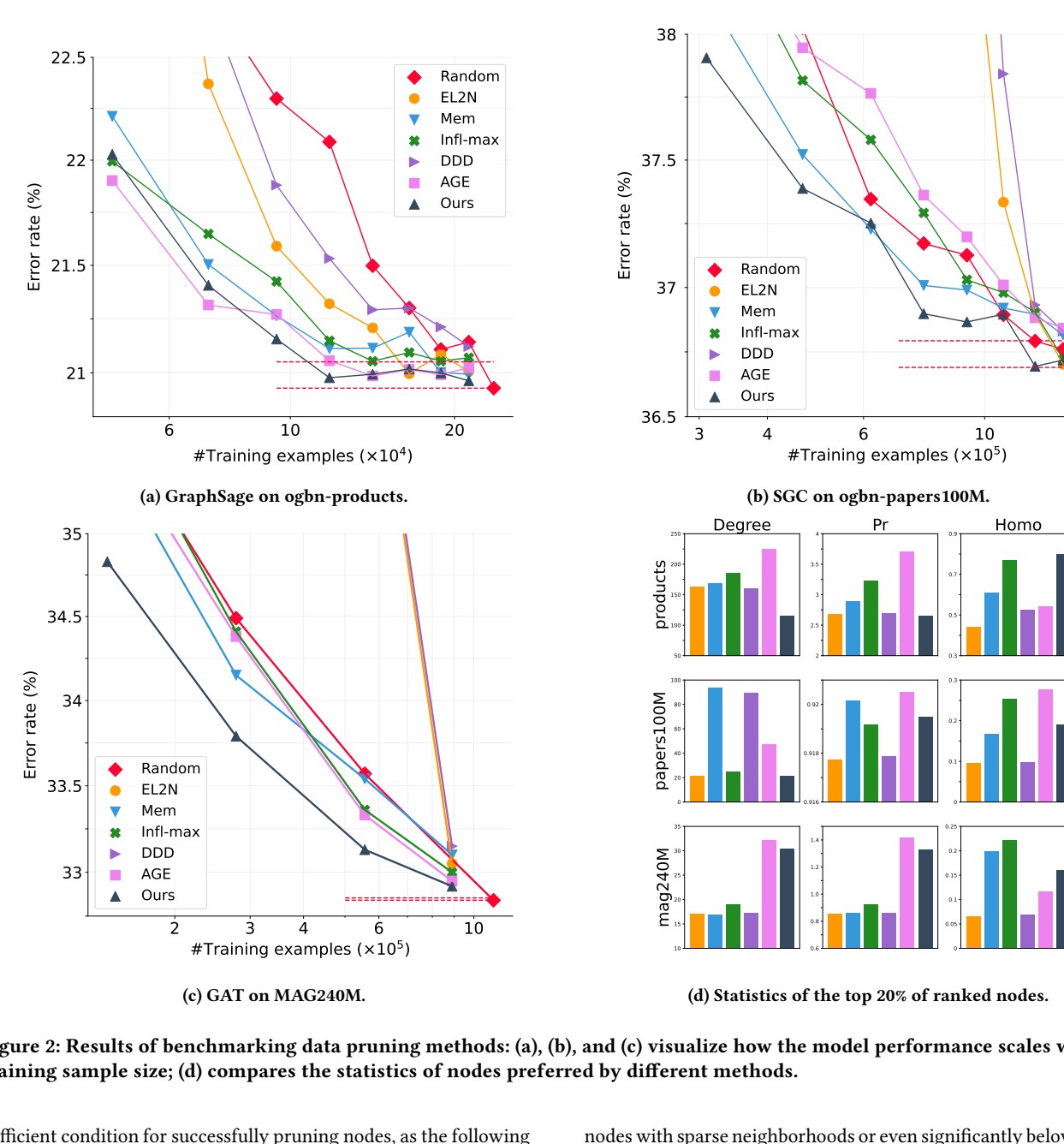

(a) GraphSage on ogbn-products.

(b) SGC on ogbn-papers100M.

(c) GAT on MAG240M.

(d) Statistics of the top 20% of ranked nodes.

**Figure 2: Results of benchmarking data pruning methods: (a), (b), and (c) visualize how the model performance scales with the training sample size; (d) compares the statistics of nodes preferred by different methods.**

sufficient condition for successfully pruning nodes, as the following finding explains.

**Failed pruning lacks the consideration of representativeness.** Despite the technical difference in how to reflect a training example's hardness, EL2N and DDD also prioritize nodes with heterophily, where the homophilic level of their top 20% of nodes is even lower than methods that have successfully beat random pruning. However, when the fraction of reserved training nodes is very limited, EL2N and DDD are remarkably surpassed by random pruning on ogbn-papers100M and MAG240M.

As GNN is essentially based on the message-passing paradigm, some studies [19, 21] have revealed its shortcomings in handling

nodes with sparse neighborhoods or even significantly below-average degrees. Thus, we conjecture that such nodes are over-rated by EL2N and DDD due to their hardness to GNNs, and those properties make such nodes less representative of the whole node set, leading to biased GNN and poor performance. Actually, Figure 2d shows that, on both ogbn-papers100M and MAG240M, EL2N and DDD's mean values of PR are the smallest two among all adopted methods, not to say that those nodes top-ranked by EL2N have very small degrees. In a word, only prioritizing hardness is inadequate, and being representative of the entire training set is necessary for successful pruning.

**Distribution shift matters**. As we have pointed out in Section 1, the semi-supervised setting of node classification tasks allows data pruning methods to utilize the features of test nodes, which, intuitively speaking, is helpful for mitigating the distribution shift issue. From Figure 2a, 2b, and 2c, we find that our method consistently outperforms baselines in the sense that it can achieve an error rate that is no worse than that of a complete set by one standard deviation, by the least number of reserved training nodes. Moreover, our method has achieved the best error rate at most of the considered pruning rates. As we will elaborate in the next section, the most salient characteristic distinguish our method from compared ones is its exploitation of the test nodes, which encourages subsets with fewer distribution shifts.

**Remark 1.** When we consolidate these findings, a test set-targeted hard-and-representative principle emerges. Specifically, under the semi-supervised setting, plausible subsets are supposed to be hard and representative for the entire training set, where measuring hardness and representativeness regarding the given test set would be more effective.

## 5 SEMI-SUPERVISED NODE PRUNING

### 5.1 Modelling

To utilize the semi-supervised setting for mitigating distribution shift, we propose to prune training nodes by prioritizing those that are more similar to testing nodes. As the first step, we need to determine how to represent the nodes and measure their similarities or distances. Inspired by recent theoretical results justifying the discrimination advantages of GNN's message propagation [16], we decided to use $H = (\tilde{D}^{-1}\tilde{A})^d X$ for representing nodes, where $d$ controls the size of the receptive field. Although nontrivial choice (i.e., $d >= 1$) has been proven to exist, guaranteeing $H$ is more discriminative than $X$, large values for $d$ tend to cause the over-smoothing issue [35]. For simplicity, we just set $d$ to be the depth of the GNNs adopted in our benchmark (3, 3, and 2 on those three datasets, respectively). The exploration of various propagation matrices and choices for $d$ is deferred to our future work.

At first glance, such a representation seems inconsistent with the representation learning functionality of GNNs since each GNN layer, including that used by GraphSage and GAT, aggregates messages with parametric transformation rather than identity mapping. However, a fairness-focused study [23] has shown that when the distance between a subgroup of test nodes and the fixed training set is measured based on such a representation, the error rate on different subgroups positively correlates with their distances to the common training set. Meanwhile, our definition of $H$ coincides with a feature propagation-based pre-processing step to accelerate GNN training on large-scale graphs [2], in which case, there would be no extra overhead to compute $H$.

When the node is well represented, the next step is to define the distance between a subset of training nodes and the given test set such that picking training nodes can be formulated as minimizing such a distance. Inspired by [23], we define the bottleneck distance between a subset $S \subseteq \mathcal{V}^{(\text{tr})}$ and the test set $\mathcal{V}^{(\text{ts})}$ as follows:

**Definition 5.1** (Bottleneck distance).
$$\Delta(S) := \max_{i \in \mathcal{V}^{(\text{ts})}} \min_{j \in S} \|H_j - H_i\|_2$$

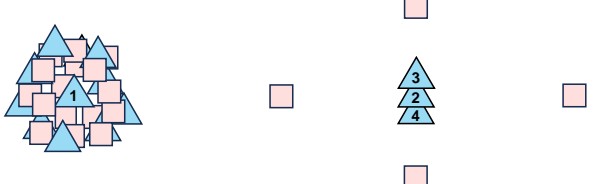

**Figure 3: A toy example with 2-dimensional node representations, where triangles represent training nodes while squares represent test nodes.**

Then we propose to formulate data pruning for node classification as a facility location problem, or more specifically, a variant of $k$-center problem. With a specified pruning rate, or equivalently, a specified number of training nodes to pick (denoted by $k$), we can determine a subset of training nodes $S_k$ by solving:

$$\min_{S} \Delta(S) \quad \text{s.t.,} \quad |S| = k. \tag{1}$$

Intuitively, each training node and test node play the role of a supplier and customer, respectively. Every customer's demand is just 1, and the volume each supplier can serve is infinite. The transportation cost per unit serviced from supplier $j$ to customer $i$ is the Euclidean distance between their corresponding node representations. We are tasked to pick at most $k$ suppliers as facilities to service all the customers so that the maximum value of transportation cost from a customer to its closest facility is minimized.

**Hardness**. By picking training node(s) that have similar representation to the test nodes, our strategy accounts for the distribution shift. In other words, when we hypothesize that the hardness of a test node can be measured by how far its closest training node is, our method can be interpreted as picking training nodes by reducing the hardness of the given test set.

**Representativeness**. One limitation of our formulation is that representativeness seems to be ignored. Considering the case shown in Figure 3, the optimal solution to Equation 1 with $k = 4$ is to pick the four numbered triangles (i.e., training nodes). Although this solution minimizes the bottleneck distance, no one would regard it as a representative subset of the complete training/test set. The bias of this solution is rooted in overemphasis on the outliers, which leads to more assignments to them ($v_1, v_2$, and $v_3$ serve the four outliers), while just $v_1$ serves the majority of test nodes. It is straight to encourage a more representative subset by uniformly restricting each supplier's supply capacity, yet solving our variant of the $k$-center problem with more constraints becomes more challenging.

### 5.2 Approximation Problem Solving

There is a well-known approximation algorithm to solve the original $k$-center problem (no separation of training and test sets but just pick facilities from one set of points), which, in each iteration, greedily picks the point most distant from the current set of facilities and adds it as another facility [7]. Inspired by this algorithm, we propose a greedy max-min selection algorithm to solve the problem defined in Equation 1, which, in the meantime, incorporates our goal of encouraging a more representative solution.

---

**Algorithm 1:** Greedy max-min selection algorithm.

**Input:** Training nodes $\mathcal{V}^{(\text{tr})}$, testing nodes $\mathcal{V}^{(\text{ts})}$, representations of training nodes $H^{(\text{tr})}_{n \times f}$, representations of testing nodes $H^{(\text{ts})}_{m \times f}$, a specified patience $T > 0$, and optionally a centrality measurement $c : \mathcal{V} \to \mathfrak{R}$ (e.g., degree or page rank).

**Output:** Ranking list of training nodes $\mathbf{r}$.

1   Initialize $\mathbf{r} = (v, )$, where $v = \arg\max_{v' \in \mathcal{V}^{(\text{tr})}} c(v')$ if $c$ is given or $v \sim \text{Uniform}(\mathcal{V}^{(\text{tr})})$, array $\mathbf{d}$ and $\mathbf{a}$ s.t., $\mathbf{d}_i = \|H^{(\text{tr})}_v - H^{(\text{ts})}_i\|_2$, and $\mathbf{a}_i = 1$ for $i = 1, \ldots, m$, and no-improvement counter by $cnt = 0$;

2   **while** $len(\mathbf{r}) < n$ **do**

3     Solve $u = \arg\max_{i \in \{1,\ldots,m\}} \mathbf{a}_i \mathbf{d}_i$ and let $q = \mathbf{d}_u$;

4     **if** $\mathbf{a}_u \neq 1$ **then**

5       Re-initialize $\mathbf{a}$ s.t., $\mathbf{a}_i = 1, i = 1, \ldots, m$;

6       **continue**

7     **else**

8       Set $\mathbf{a}_u = 0$;

9     **end**

10    Solve $v = \arg\min_{v' \in \mathcal{V}^{(\text{tr})} \wedge v' \notin \mathbf{r}} \|H^{(\text{tr})}_{v'} - H^{(\text{ts})}_u\|_2$ and let $s = \|H^{(\text{tr})}_v - H^{(\text{ts})}_u\|_2$;

11    **if** $s < q$ **then**

12      Append $v$ to the end of $\mathbf{r}$;

13      Update $\mathbf{d}$ s.t., $\mathbf{d}_i = \min(\mathbf{d}_i, \|H^{(\text{tr})}_v - H^{(\text{ts})}_i\|_2), i = 1, \ldots, m$;

14      Set $cnt = 0$;

15    **else**

16      Update $cnt = cnt + 1$;

17      **if** $cnt \geq T$ **then**

18        Re-initialize $\mathbf{a}$ s.t., $\mathbf{a}_i = 1, i = 1, \ldots, m$;

19        $v = \arg\max_{v' \in \mathcal{V}^{(\text{tr})} \wedge v' \notin \mathbf{r}} c(v')$ if $c$ is given or $v \sim \text{Uniform}(\mathcal{V}^{(\text{tr})} \setminus \mathbf{r})$;

20        Append $v$ to the end of $\mathbf{r}$;

21        Re-initialize $\mathbf{d}$ s.t., $\mathbf{d}_i = \|H^{(\text{tr})}_v - H^{(\text{ts})}_i\|_2, i = 1, \ldots, m$;

22        Set $cnt = 0$;

23      **end**

24    **end**

25 **end**

---

We present the pseudo-code in Algorithm 1. Overall, our algorithm is also a greedy algorithm, which, in each iteration, relies on the test node, whose representation has the maximal distance from currently selected training nodes, as a reference to pick the training node with the minimal distance to it. However, it should be aware that the separation of training and test sets makes our variant quite different from that dedicated to the original $k$-center problem. In each iteration, we restrict the candidates to training nodes that have not been included as a facility, as the original greedy algorithm does. However, we encounter extra design choices, e.g., whether a test node can be repeatedly used as a reference.

To mitigate the overemphasis on outlier(s), we maintain a flag (i.e., $\mathbf{a}_i$) for each test node and deactivate it once the test node is used as the reference in an iteration. Considering the toy example shown in Figure 3, suppose the initial facility set is $\{v_1\}$, then the reference would be the rightmost test node. Without maintaining the flags, after adding $v_2$ to the facility set, the reference in the following iterations is still that test node, which would suggest $v_3$ and $v_4$ as facility, making the selected subset biased toward it.

Besides, our algorithm offers a hyperparameter $T$ named "patience". Once this number of consecutive iterations has not found a facility that can effectively reduce the distance from the reference to the set of facilities (Line 17), we would re-activate all test nodes (Line 18) and re-initialize their current distances to the facility set (Line 21). Considering the toy example in Figure 3, in the second iteration, the current facility set is $\{v_1, v_2\}$, and thus the reference is the test node in the middle of the figure, which has no facility that can further reduce its distance to the facility set, namely, the "no-improvement counter" will be increased by one (Line 16). With our designed mechanism and suppose $T = 1$, we have a large probability of randomly adding one training node from the majority at the left-hand side as a facility. Otherwise, we would add either $v_3$ or $v_4$, overemphasizing the outliers.

Algorithm 1 has an $O(n \times m)$ complexity in terms of distance calculation. However, we can effortlessly change it into a mini-batch mode and utilize the parallel processing capacity of GPU. Hence, its overhead is often lower than training a model, which some related methods require.

## 5.3 Results and Analysis

We evaluate our method based on our benchmark so that it is rigorously and fairly compared with related baselines. On ogbn-products (see Figure 2a), the model trained on just half of the training nodes picked by our method can achieve an error rate no worse than that of the complete set by a standard deviation, namely, half data can lead to a statistically comparable performance by our method. On ogbn-papers100M, (see Figure 2b), our method is again the first to achieve an averaged error rate no larger than that of the complete set plus a standard deviation. Besides, with the top 80% of training nodes ranked by our method, we can achieve an averaged error rate almost the same as that of the complete set. On MAG240M (see Figure 2c), our method outperforms baselines by a remarkable margin at all considered pruning rates.

To better understand the advantages of our method, the first question is how to derive some performance (e.g., hardness) guarantee for the subsets determined by our approximation algorithm. Then, more importantly, the main question is how to relate a GNN's generalization risk on test nodes to the hardness and representativeness of the subset of training nodes picked by our method.

*5.3.1 Analysis of hardness reduction.* Before presenting the performance guarantee for the picked subsets, we define bottleneck distance (see Definition 5.1)'s counterpart for the training set itself as $\Delta'(\mathcal{S}) := \max_{i \in \mathcal{V}^{(\text{tr})}} \min_{j \in \mathcal{S}} \|H_j - H_i\|_2$, which reflects how well the selected facilities service the training nodes (as customers). Suppose $\mathcal{S}$ is a subset produced by Algorithm 1. Let $\mathcal{S}'^*$ denote the optimum set of facilities that minimizes the bottleneck distance for the training set and have the same cardinality as $\mathcal{S}$ (i.e., $|\mathcal{S}'^*| = |\mathcal{S}|$).

We offer a guarantee for $\Delta(\mathcal{S})$ to show how well the solution Algorithm 1 seeks is, which can be formally stated as follows:

**Proposition 1.** $\Delta(\mathcal{S}) \leq 2\Delta'(\mathcal{S}'^*) + \Delta(\mathcal{V}^{(\text{tr})})$.

Due to the limited space, we defer our proof to Appendix B. As discussed in Section 5.1, when we interpret the distance from a test node to the closest training node from a picked subset as a measure of this test node's hardness, reducing the bottleneck distance means making the test set easier. Thus, Proposition 1 guarantees that the picked subset would not let the hardness of the test set exceed an upper bound. We realize this is not a tight bound and thus empirically compare our picked subsets to those constructed by applying the classical greedy algorithm to allocate facilities for serving the training nodes themselves. The results are presented in Figure 4, where the $x$-axis represents the number of picked training nodes (i.e., $k$), and the $y$-axis represents the averaged minimum distance over the next $k$ referenced test nodes. Except for ogbn-products, our algorithm leads to a significantly faster decrease in such distance as the $k$ increases, which can be interpreted as being more effective in eliminating the distribution shift and thus reducing the test set's hardness. Both algorithms behave almost the same on ogbn-products because there are nearly ten times the test nodes of the training nodes.

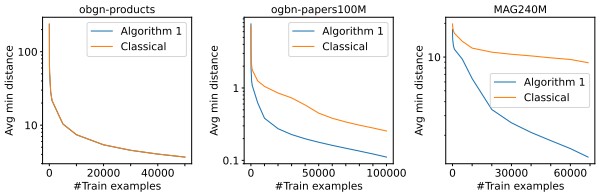

**Figure 4: Comparison of Algorithm 1 and classical greedy algorithm: effectiveness in reducing test set hardness.**

As discussed in Section 5.1, intuitively speaking, a more uniform assignment of facilities (i.e., picked training nodes) to test nodes corresponds to a more representative subset for the test set. However, as no such notion is directly optimized in Algorithm 1, we cannot offer any guarantee for the representativeness of $\mathcal{S}$ by now and leave it as our future work.

*5.3.2 Generalization risk analysis for pruned training data.* Our analysis is primarily built upon a PAC-Bayesian framework introduced for analyzing the performance fairness among different subgroups of test nodes [23]. The most salient differences are two-fold: (1) We are interested in models learned from our picked subsets rather than a fixed training set, which requires changing some assumptions to a different form and restricts the range of applicable pruning rate; (2) More importantly, we eliminate their assumption of uniform assignment but instead introduce a novel notion and use it to account for the representativeness of picked subsets in the generalization risk bound.

**Settings**. Data are generated following a process wherein an underlying aggregation function $g$, such as a $L$-layer GNN, produces $Z_{|\mathcal{V}|\times f}$, and node labels are conditional independent given $Z$, namely, each node label is independently sampled from an unknown conditional probability distribution $y_i \sim P(y|Z_i)$. This is

not conflict with our setting that node labels are not i.i.d., because $Z_i$s are obviously allowed to be non-i.i.d. This is also consistent with the existence of distribution shift because $Z_{1:n}$ and $Z_{n+1:n+m}$ may obey different distribution.

To analyze our data pruning method, it is straightforward to assume $Z$ coincides with $H$ used as input to Algorithm 1. Nonetheless, to better understand why our simple random walk-based aggregation works consistently well, we analyze its difference against a more realistic $g$: Suppose $g$ has feature transformation in each step of message propagation but no non-linearity, then $\exists W$ such that, $Z = g(X, A) = (\tilde{D}^{-1}\tilde{A})^d XW_1 \cdots W_l = (\tilde{D}^{-1}\tilde{A})^d XW = HW$. As a result, $\forall i, j, \|Z_i - Z_j\|_2^2 = (H_i - H_j)(WW^{\text{T}})(H_i - H_j)^{\text{T}} \leq \lambda_{\max}\|H_i - H_j\|_2^2$, where $\lambda_{\max} \geq 0$ is the largest eigenvalue of $WW^{\text{T}}$. Thus, for a subset of training node $\mathcal{S}$, $\max_{i\in\mathcal{V}^{(\text{ts})}} \min_{j\in\mathcal{S}} \|Z_i - Z_j\|_2 \leq \sqrt{\lambda_{\max}}\Delta(\mathcal{S})$. Based on the above analysis, the first implication is why feature normalization is often helpful in practice. The second implication is that it is reasonable to skip the difference of just a scaling factor (i.e., $\sqrt{\lambda_{\max}}$). Thus, in the following analysis, we simplify this data generation process by letting $Z = H$ and use them interchangeably.

Here, each classifier drawn from the hypothesis space $\mathcal{H}$ is ReLU-activated $L$-layer Multi-layer perceptron (MLP) with $\{W_l\}_{l=1}^{L}$ as model parameters. Then each $h \in \mathcal{H}$ is applied to $H$ to produce predictions $\hat{Y} = h(H; \{W_l\}_{l=1}^{L})$. This $\mathcal{H}$ seems quite restricted, but both the widely adopted SGC [34] and the theoretical result that $H$ contains rich information [16] confirm that it is a reasonable choice for analyzing our method. The largest width of all the MLP layers is denoted by $b$.

Recall the margin loss we have introduced in Section 2, then the empirical margin loss of any classifier $h \in \mathcal{H}$ on a node set $\mathcal{S}$ can be defined as $\hat{\mathcal{L}}_{\mathcal{S}}^{\gamma}(h) := \frac{1}{|\mathcal{S}|} \sum_{i\in\mathcal{S}} l^{\gamma}(\hat{Y}_i, y_i)$, where $\hat{Y}$ is the predictions made by classifier $h$. Naturally, the expected margin loss is defined as $\mathcal{L}_{\mathcal{S}}^{\gamma}(h) := \mathbb{E}_{y_i \sim P(y|Z_i),\ i\in\mathcal{S}}[\hat{\mathcal{L}}_{\mathcal{S}}^{\gamma}(h)]$.

**Analysis**. Conventionally, our analysis starts with a widely adopted smoothness assumption:

**Assumption 5.1** (Smoothness). *For each $j = 1, \ldots, |\mathcal{Y}|$, there is a $\eta_j : \mathfrak{R}^h \to \mathfrak{R}$ such that $\eta_j(Z_i) = P(y = j|Z_i)$, and all these $\eta_j$s are $c$-Lipschitz continuous function.*

Then, the essential difference between our analysis and that dedicated to performance fairness comes. That work bounds, for a fixed classifier, the difference between the expected margin loss on any subgroup of test nodes and that on their common training set, with another strong and unrealistic assumption that each training node has disjoint and equal-sized near set (a near set consists of test nodes within the bottleneck distance to that training node). Instead, we have gained knowledge from our benchmark (see Section 4.2) that whether a selected subset is representative of the test set is crucial to the model performance, where, regarding Algorithm 1, the representativeness can be reflected by the uniformness of facility assignment. Thus, we aim to characterize how the uniformness of facility assignment ultimately affects the generalization ability of a classifier trained on the selected subset. For such purpose, we define the max flow of a selected subset as follows:

**Definition 5.2** (Max flow of a selected subset). Given a subset of training nodes $\mathcal{S}$, construct a bi-partite graph for it, where a set of

$m$ nodes corresponding to the original test nodes are connected to a set of $|\mathcal{S}|$ nodes corresponding to the selected training nodes. There is a directed edge from a node corresponding to $i \in \mathcal{V}^{(ts)}$ to a node corresponding to a selected training node $j \in \mathcal{S}$ if $\|H_i - H_j\|_2 \leq \Delta(\mathcal{S})$, and the edge weight is $+\infty$. Then a source node is connected to all the $m$ nodes by directed edges with the same weight $\frac{|\mathcal{S}|}{m}$, and all those $|\mathcal{S}|$ nodes are connected to a target node by directed edges with the same weight 1. The maximum feasible flow from the source to the target is said to be the max flow of $\mathcal{S}$, i.e., $flow(\mathcal{S})$.

In the Appendix, we provide readers with an illustrative example of such a bi-partite graph (see Figure 7).

Based on this novel notion, we can bound the difference between the expected margin loss on the test set and that on a subset of training nodes $\mathcal{S}$ for a fixed classifier as follows:

**Lemma 2.** For a $L$-layer classifier $h \in \mathcal{H}$ with parameters $W_1, \ldots, W_L$, define $T_h := \max_{l=1,\ldots,L} \|W_l\|_2$. Under Assumption 5.1, suppose $\mathcal{S}$ is a subset of training nodes, for any $\gamma \geq 0$, if $\Delta(\mathcal{S})T_h^L \leq \frac{\gamma}{4}$, then

$$\mathcal{L}_{\mathcal{V}^{(ts)}}^{\gamma/2}(h) - \mathcal{L}_{\mathcal{S}}^{\gamma}(h) \leq \frac{2(|\mathcal{S}|-flow(\mathcal{S}))}{|\mathcal{S}|} + |\mathcal{Y}|c\Delta(\mathcal{S}).$$

Due to the limited space, we defer our proof to Appendix B.

**Remark 2.** As can be seen, without Assumption 2 of [23], the first term in our upper bound cannot be eliminated. Instead, we consider a more general case and account for the factor of representativeness characterized by the defined max flow.

Then, the remaining path toward our ultimate theoretical result follows the same rationale as that in [23]. As our focus is the generalization risk of different subsets of training nodes rather than different subgroups of test nodes, we no longer need to index different subgroups yet use $\mathcal{S}_k, k \leq n$, to index the subsets produced by our algorithm, where $|\mathcal{S}_k| = k$. It is worth noticing that $k$ is not allowed to be too small. In practice, too little $k$ often causes performance corruption. Moreover, our theoretical result requires that $|\mathcal{S}|$ be large enough, which is essentially demanded by the following assumption. Specifically, due to the difference in purpose and our upper bound (see Lemma 2) against theirs, our counterpart of Assumption 3 in [23] needs to be changed as follows:

**Assumption 5.2** (Small generalization discrepancy). *For any of our considered subset $\mathcal{S}_k$, let $P$ be a distribution on $\mathcal{H}$, which is defined by sampling the vectorized MLP parameters from $\mathcal{N}(0, \sigma^2 I)$ for some $\sigma^2 \leq \frac{(\gamma/8\Delta(\mathcal{S}_k))^{2/L}}{2b(\lambda k^{-\alpha}+\ln 2bL)}$. For any $L$-layer classifier $h \in \mathcal{H}$ with model parameters $W_1^h, \ldots, W_L^h$, again define $T_h := \max_{l=1,\ldots,L} \|W_l^h\|_2$. Assume there exists some $0 < \alpha < \frac{1}{4}$ such that:*

$$P_{h \sim \mathcal{H}}\Big(\mathcal{L}_{\mathcal{V}^{(ts)}}^{\gamma/4}(h) - \mathcal{L}_{\mathcal{S}_k}^{\gamma/2}(h) > k^{-\alpha} + \frac{2(k-flow(\mathcal{S}_k))}{k}$$
$$+ |\mathcal{Y}|c\Delta(\mathcal{S}_k) \quad |T_h^L\Delta(\mathcal{S}_k) > \frac{\gamma}{8}\Big) \leq e^{-k^{2\alpha}}.$$

Similarly, Assumption 4 in [23] should be changed according to our cases as follows:

**Assumption 5.3** (Constant norms). *For any of our considered subset $\mathcal{S}_k$, define $B_k := \max_{i \in \mathcal{V}^{(ts)} \cup \mathcal{S}_k} \|H_i\|_2$. For any classifier $\tilde{h} \in \mathcal{H}$ with parameters $\{\tilde{W}_l\}_{l=1}^L$, assume $\forall l, \|\tilde{W}_l\|_F \leq C$. Assume $B_k$ and $C$ are constants no matter which value $k$ takes.*

Finally, combining all the above assumptions and intermediate results, we can reach our ultimate theoretical result:

**Theorem 3.** *For any $\tilde{h} \in \mathcal{H}$ with parameters $\{\tilde{W}_l\}_{l=1}^L$. Under Assumptions 5.1, 5.2, and 5.3, for any of our considered subset $\mathcal{S}_k$ that ensures $k$ large enough, for any $\gamma \geq 0$, with probability at least $1 - \delta$ over the sample of $y_i, i \in \mathcal{S}_k$, we have*

$$\mathcal{L}_{\mathcal{V}^{(ts)}}^0(\tilde{h}) \leq \hat{\mathcal{L}}_{\mathcal{S}_k}^{\gamma}(\tilde{h}) + O\Big(\frac{b\sum_{l=1}^L \|\tilde{W}_l\|_F^2}{k^\alpha(\gamma/8)^{2/L}}\Delta(\mathcal{S}_k)^{2/L} + $$
$$\frac{1}{k^{2\alpha}}\ln\frac{LC(2B_k)^{1/L}}{\gamma^{1/L}\delta} + \frac{1}{k^{1-2\alpha}} + $$
$$\frac{(k-flow(\mathcal{S}_k))}{k} + |\mathcal{Y}|c\Delta(\mathcal{S}_k)\Big).$$

The complete analysis, including more detailed intermediate results and the proof of this theorem, can be found in Appendix B.

## 5.4 Sensitivity Analysis

When the data distribution occasionally causes Algorithm 1 to encounter no-improvement events, the patience $T$ would matter. Thus, we conduct sensitivity analysis by executing Algorithm 1 on ogbn-papers100M, with different choices of $T \in \{1, 3, 5, 15, 45\}$, and comparing their resulting ranking list of training nodes. For each ranking list, how model performance scales with the number of reserved training nodes is shown in Figure 5. Whichever $T$ is, the resulting ranking list can beat random pruning. Besides, some choices (e.g., 3 and 5) can perform better than what we used in our main experiment (i.e., 15) at the pruning rates of 60% and 80%.

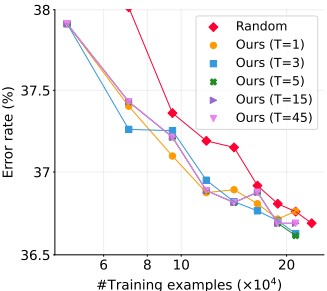

**Figure 5: Comparison of different choices of patience $T$.**

## 6 CONCLUSIONS AND FUTURE DIRECTIONS

Through extensive studies on large-scale graphs, we reveal an unsatisfactory power law scaling for training GNN to classify nodes, beat such a law by several data pruning methods, and attain a principle for picking useful training nodes. Based on this test set-targeted hard-and-representative principle, we not only design a novel method that can outperform existing methods but also present a theoretical analysis accounting for both hardness and representativeness factors. Being a promising way to reduce the overhead of GNN training, our study also offers the community more insights into the relation between training samples and model performance in node classification tasks.

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

## A IMPLEMENTATION DETAILS

**GNN models**. We consider GraphSage, SGC, and GAT in our benchmark, where all their implementations are based on the PyG version of model implementations provided by OGB team. For the generality, we keep the default hyper-parameters unchanged. Readers who are interested in the details are referred to this GitHub repository.

**Benchmark suite**. Our setting largely follows that in [30]. For each data pruning method, we first apply it to generate a ranking list of all the training nodes. Then, to evaluate this method at a certain pruning rate, or equivalently, at a certain number of reserved training nodes (i.e., $k$), we first randomly sample a fraction of $\frac{k}{2n}$ training nodes of each class. This is needed to alleviate the influence of label distribution discrepancy caused by data pruning. Then the remaining $\frac{k}{2}$ training nodes are determined by the top-$\frac{k}{2}$ elements in the ranking list that have not been included.

Then a specified GNN model is learned from this picked subset and evaluated, where all the training and evaluation procedures are the same across different data pruning methods. There are two sources of randomness: one is the half of training nodes randomly selected, the other is the randomness in model training stage. Thus, on each dataset, we marginalize the first factor by considering five different seeds. Each seed corresponds to a specific subset of the considered data pruning method. Then, on each subset, we repeat the model training and evaluation procedure for 10, 5, 1 times on ogbn-products, ogbn-papers100M, and MAG240M, respectively. Finally, we average the reported error rate on test set, where the corresponding model checkpoint is the one achieves the best validation performance.

## B FULL THEORETICAL ANALYSIS

### B.1 Analysis of our approximation algorithm

We re-state Proposition 1 as follows:

**Proposition 4.** $\Delta(S) \leq 2\Delta'(S'^*) + \Delta(\mathcal{V}^{(\mathrm{tr})})$.

PROOF. As $S$ is constructed by iteratively adding $s_1, s_2, \ldots, s_k$ into it, we use $t_1, t_2, \ldots, t_k$ to denote the reference point (i.e., the test node with maximal distance to the current facility set) at each iteration. Meanwhile, we use $s_{k+1}$ to denote the next training node being included if we continue the procedure.

Let $\Delta_i, i = 1, \ldots, k$ denote the bottleneck distance for the subset consisting of the first $i$ elements added to $S$. Then it is obvious that $\Delta_{i+1} \leq \Delta_i$ because the distance of each testing node to its closest facility will either decrease or at least be unchanged.

Next, we show that $\forall j < j' \leq k + 1, \|H_{s_j}^{(\mathrm{tr})} - H_{s_{j'}}^{(\mathrm{tr})}\| \geq \Delta_{j'-1} - \Delta(\mathcal{V}^{(\mathrm{tr})})$. Actually, this proposition can be restated as $\forall i = 2, \ldots, k+1, \forall j, j' \leq i (j \neq j'), \|H_{s_j}^{(\mathrm{tr})} - H_{s_{j'}}^{(\mathrm{tr})}\| \geq \Delta_{j'-1} - \Delta(\mathcal{V}^{(\mathrm{tr})})$. The case of $i = 2$ is obvious. $\Delta_1 = \|H_{s_1}^{(\mathrm{tr})} - H_{t_1}^{(\mathrm{ts})}\|$. $s_2$ would not be far way from $t_1$ by a distance larger than $\Delta_1$, otherwise $t_1$ will be skipped and inactivated (Line 15 in Algorithm 1). Then the worst case (i.e., the case leading to the closest $s_2$ to $s_1$) is that $s_2$ lies in the line segment between $s_1$ and $t_1$. Meanwhile, $s_2$ can not be far away from $t_1$ by $\Delta(\mathcal{V}^{(\mathrm{tr})})$ due to that otherwise there must exist a closer training node to $t_1$. Thus, the base case $\|H_{s_1}^{(\mathrm{tr})} - H_{s_2}^{(\mathrm{tr})}\| \geq \Delta_1 - \Delta(\mathcal{V}^{(\mathrm{tr})})$ has been established. By induction, we first assume the case of $i$ is right

and start from it to show the case of $i + 1$ is also right. Specifically, the closest training node (w.l.o.g., denote it by $s_j, 1 \leq j \leq i$) to $t_i$ is distant from it by $\Delta_i$, and again the existence of an improvement brought in by $s_{i+1}$ ensures that $\|H_{s_j}^{(\mathrm{tr})} - H_{s_{i+1}}^{(\mathrm{tr})}\| \geq \Delta_i + \Delta(\mathcal{V}^{(\mathrm{tr})})$. Meanwhile, as $\forall j' \neq j, \|H_{s_{j'}}^{(\mathrm{tr})} - H_{t_i}^{(\mathrm{ts})}\| \geq \Delta_i$ and $\|H_{s_{i+1}}^{(\mathrm{tr})} - H_{t_i}^{(\mathrm{ts})}\| \leq \Delta(\mathcal{V}^{(\mathrm{tr})})$, we have shown that $\forall j \leq i, \|H_{s_j}^{(\mathrm{tr})} - H_{s_{i+1}}^{(\mathrm{tr})}\| \geq \Delta_i + \Delta(\mathcal{V}^{(\mathrm{tr})})$. Thus, we complete the induction step, where the main intuition behind this induction proof is visualized in Figure 6.

Finally, for an arbitrary facility set $C$ with $\|C\| = \|S\| = k$, we will show $\Delta(S) \leq 2\Delta'(C) + \Delta(\mathcal{V}^{(\mathrm{tr})})$. Considering $S \cup \{s_{k+1}\}$, there must be at least two of its elements that lie within the distance $\Delta'(C)$ to a certain element $c \in C$ by the pigeonhole principle. Suppose these two facilities are $s_j, s'_j, j, j' \leq k+1, \|H_{s_j}^{(\mathrm{tr})} - H_{s_{j'}}^{(\mathrm{tr})}\| \geq \Delta_k - \Delta(\mathcal{V}^{(\mathrm{tr})}) = \Delta(S) - \Delta(\mathcal{V}^{(\mathrm{tr})})$ as we have just proved above. Meanwhile, by triangle inequality, $\|H_{s_j}^{(\mathrm{tr})} - H_{s_{j'}}^{(\mathrm{tr})}\| \leq \|H_{s_j}^{(\mathrm{tr})} - H_c^{(\mathrm{tr})}\| + \|H_c^{(\mathrm{tr})} - H_{s_{j'}}^{(\mathrm{tr})}\| \leq 2\Delta'(C)$. Combining these inequalities and exploiting the arbitrary of choosing $C$, we have $\Delta(S) \leq 2\Delta'(S'^*) + \Delta(\mathcal{V}^{(\mathrm{tr})})$, where $S'^*$ is the optimum set of facilities for reducing the bottleneck distance on the training set itself. □

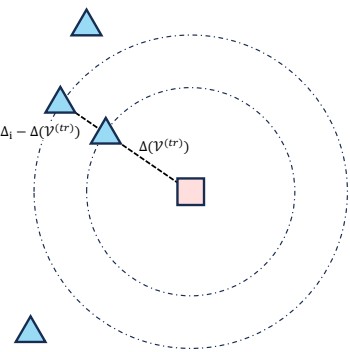

**Figure 6: Bounding the distance between next added facility to existing facilities.**

One limitation of the above analysis is that we have not considered the re-start mechanism of our propose algorithm. However, this is equivalent to consider a very large patience $T$, where our sensitivity analysis (see Section 5.4) has shown that the end-to-end performance of our data pruning method is robust to changes of $T$.

### B.2 Analysis of the performance of models learned on pruned data

For readers to better understand our novel notion for uniform assignment of facilities (recall Definition 5.2), we show such a bipartite graph in Figure 7 as an example.

We re-state our Lemma 2 as follows:

**Lemma 5.** For a $L$-layer classifier $h \in \mathcal{H}$ with parameters $W_1, \ldots, W_L$, define $T_h := \max_{l=1,\ldots,L} \|W_l\|_2$. Under Assumption 5.1, suppose $S$ is a subset of training nodes, for any $\gamma \geq 0$, if $\Delta(S) T_h^L \leq \frac{\gamma}{4}$, then $\mathcal{L}_{\mathcal{V}^{(\mathrm{ts})}}^{\gamma/2}(h) - \mathcal{L}_S^{\gamma}(h) \leq |\mathcal{Y}| c\Delta(S) + \frac{2(|S| - flow(S))}{|S|}$.

**Table 1: Statistics of adopted datasets.**

| name | #node | #edge | #feature | #class | split by | train/valid/test |
|---|---|---|---|---|---|---|
| products | 2,449,029 | 61,859,640 | 100 | 47 | species | 8%/1.6%/90.4% |
| papers100M | 111,059,956 | 1,615,685,872 | 128 | 172 | time | 78%/8.1%/13.9% |
| MAG240M | 121,751,226 | 1,296,620,606 | 768 | 153 | time | 83%/10.4%/6.6% |

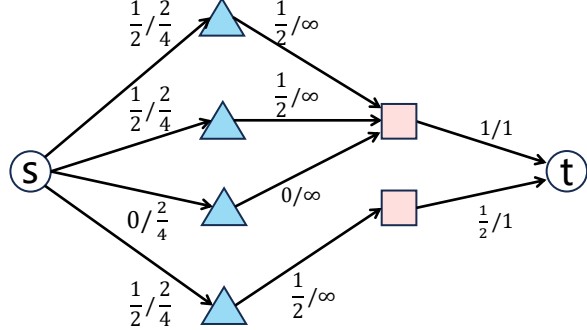

**Figure 7: Bi-partite graph construction based on $k$-center solution.**

PROOF. To keep our notation terse, we use $\eta_k(i)$ to denote $\eta_k(Z_i)$ and define $\mathcal{L}_{i,k}^{\gamma} := \mathbf{1}_{\hat{Y}_{i,k} \leq \gamma + \max_{k' \neq k} \hat{Y}_{i,k'}}$.

$$\mathcal{L}_{\mathcal{V}^{(ts)}}^{\gamma/2}(h) - \mathcal{L}_{\mathcal{S}}^{\gamma}(h) = \mathbb{E}_{y_i}\Big[\frac{1}{m}\sum_{i \in \mathcal{V}^{(ts)}} \mathcal{L}_{i,y_i}^{\gamma/2}\Big] - \mathbb{E}_{y_j}\Big[\frac{1}{|\mathcal{S}|}\sum_{j \in \mathcal{S}} \mathcal{L}_{j,y_j}^{\gamma}\Big]$$

$$= \frac{1}{|\mathcal{S}|}\mathbb{E}_{y_i,y_j}\Big[\sum_{j \in \mathcal{S}}\sum_{i \in \mathcal{V}^{(ts)}} \omega(i,j)\mathcal{L}_{i,y_i}^{\gamma/2} - \mathcal{L}_{j,y_j}^{\gamma}\Big],$$

where we let $\forall i, j, \omega(i,j) \geq 0$ and is positive only if $\|H_i^{(tr)} - H_j^{(ts)}\| \leq \Delta(\mathcal{S})$, and $\forall i \in \{n+1, \ldots, n+m\}, \sum_{j \in \mathcal{S}} \omega(i,j) = \frac{|\mathcal{S}|}{m}$.

$$\frac{1}{|\mathcal{S}|}\mathbb{E}_{y_i,y_j}\Big[\sum_{j \in \mathcal{S}}\sum_{i \in \mathcal{V}^{(ts)}} \omega(i,j)\mathcal{L}_{i,y_i}^{\gamma/2} - \mathcal{L}_{j,y_j}^{\gamma}\Big]$$

$$= \frac{1}{|\mathcal{S}|}\sum_{j \in \mathcal{S}}\Big(\sum_{i \in \mathcal{V}^{(ts)}} \omega(i,j)\sum_{k=1}^{|\mathcal{Y}|}\eta_k(i)\mathcal{L}_{i,k}^{\gamma/2} - \sum_{k=1}^{|\mathcal{Y}|}\eta_k(j)\mathcal{L}_{j,k}^{\gamma}\Big)$$

$$= \frac{1}{|\mathcal{S}|}\sum_{j \in \mathcal{S}}\Big(\sum_{i \in \mathcal{V}^{(ts)}} \omega(i,j)\sum_{k=1}^{|\mathcal{Y}|}\eta_k(i)(\mathcal{L}_{i,k}^{\gamma/2} - \mathcal{L}_{j,k}^{\gamma})$$

$$+ \sum_{i \in \mathcal{V}^{(ts)}} \omega(i,j)\sum_{k=1}^{|\mathcal{Y}|}\eta_k(i)\mathcal{L}_{j,k}^{\gamma} - \sum_{k=1}^{|\mathcal{Y}|}\eta_k(j)\mathcal{L}_{j,k}^{\gamma}\Big)$$

$$\leq \frac{1}{|\mathcal{S}|}\sum_{j \in \mathcal{S}}\Big(\sum_{i \in \mathcal{V}^{(ts)}} \omega(i,j)\sum_{k=1}^{|\mathcal{Y}|}\eta_k(i)\mathcal{L}_{j,k}^{\gamma} - \sum_{k=1}^{|\mathcal{Y}|}\eta_k(j)\mathcal{L}_{j,k}^{\gamma}\Big),$$

where the last inequality is valid since $\forall k \in \{1, \ldots, |\mathcal{Y}|\}, \forall i \in \mathcal{V}^{(ts)}, j \in \mathcal{S}, \mathcal{L}_{i,k}^{\gamma/2} \leq \mathcal{L}_{i,k}^{\gamma}$ due to $\|\hat{Y}_i - \hat{Y}_j\|_{\infty} \leq \|H_i - H_j\|_2 \prod_{l=1}^{L}\|W_l\|_2 \leq \Delta(\mathcal{S})T_h^L \leq \frac{\gamma}{4}$, as the proof of Lemma 5 in [23] has shown.

$$\frac{1}{|\mathcal{S}|}\sum_{j \in \mathcal{S}}\Big(\sum_{i \in \mathcal{V}^{(ts)}} \omega(i,j)\sum_{k=1}^{|\mathcal{Y}|}\eta_k(i)\mathcal{L}_{j,k}^{\gamma} - \sum_{k=1}^{|\mathcal{Y}|}\eta_k(j)\mathcal{L}_{j,k}^{\gamma}\Big)$$

$$= \frac{1}{|\mathcal{S}|}\sum_{j \in \mathcal{S}}\sum_{i \in \mathcal{V}^{(ts)}}\Big(\omega(i,j)\sum_{k=1}^{|\mathcal{Y}|}\eta_k(i)\mathcal{L}_{j,k}^{\gamma} - \omega'(i,j)\sum_{k=1}^{|\mathcal{Y}|}\eta_k(j)\mathcal{L}_{j,k}^{\gamma}\Big)$$

$$= \frac{1}{|\mathcal{S}|}\sum_{j \in \mathcal{S}}\sum_{i \in \mathcal{V}^{(ts)}}\sum_{k=1}^{|\mathcal{Y}|}\mathcal{L}_{j,k}^{\gamma}\Big(\omega(i,j)\eta_k(i) - \omega'(i,j)\eta_k(j)\Big)$$

$$= \frac{1}{|\mathcal{S}|}\sum_{j \in \mathcal{S}}\sum_{i \in \mathcal{V}^{(ts)}}\sum_{k=1}^{|\mathcal{Y}|}\mathcal{L}_{j,k}^{\gamma}\Big((\omega(i,j) - \omega'(i,j))\eta_k(i)$$

$$+ \omega'(i,j)(\eta_k(i) - \eta_k(j))\Big)$$

$$\leq \frac{1}{|\mathcal{S}|}\sum_{j \in \mathcal{S}}\sum_{i \in \mathcal{V}^{(ts)}}\sum_{k=1}^{|\mathcal{Y}|}\mathcal{L}_{j,k}^{\gamma}(\omega(i,j) - \omega'(i,j))\eta_k(i) + |\mathcal{Y}|c\Delta(\mathcal{S}),$$

where the last inequality comes from that $\forall k \in \{1, \ldots, |\mathcal{Y}|\}, \forall i \in \mathcal{V}^{(ts)}, j \in \mathcal{S}, \eta_k(i) - \eta_k(j) \leq \|H_i - H_j\| \leq c\Delta(\mathcal{S})$, and $\forall i, j, \omega'(i,j) \geq 0$ and is positive only if $\|H_i - H_j\| \leq \Delta(\mathcal{S})$, and $\forall j, \sum_{i \in \mathcal{V}^{(ts)}} \omega'(i,j) = 1$.

The remaining difficulty is to bound the first term:

$$\frac{1}{|\mathcal{S}|}\sum_{j \in \mathcal{S}}\sum_{i \in \mathcal{V}^{(ts)}}\sum_{k=1}^{|\mathcal{Y}|}\mathcal{L}_{j,k}^{\gamma}(\omega(i,j) - \omega'(i,j))\eta_k(i)$$

$$\leq \frac{1}{|\mathcal{S}|}\sum_{j \in \mathcal{S}}\sum_{i \in \mathcal{V}^{(ts)}}|\omega(i,j) - \omega'(i,j)|$$

$$(\text{as } 0 \leq \mathcal{L}_{j,k}^{\gamma} \leq 1, 0 \leq \eta_k(i), \text{ and } \sum_k \eta_k(i) = 1),$$

where the RHS is minimized by adjusting $\omega(i,j)$ and $\omega'(i,j)$ in a way that "mostly reducing the wasted flow" in the flow network corresponding to $\mathcal{S}$ as Definition 5.2 introduced. Specifically, the RHS can be as small as $\frac{2(|\mathcal{S}|-flow(\mathcal{S}))}{|\mathcal{S}|}$, where no matter $\omega(i,j) < \omega'(i,j)$ or $\omega(i,j) > \omega'(i,j)$, it means a waste happened either at an edge from the source node to the test node $i$ or at an edge from the training node $j$ to the target node.

Combining all the above results, we know that $\mathcal{L}_{\mathcal{V}^{(ts)}}^{\gamma/2}(f) - \mathcal{L}_{\mathcal{S}}^{\gamma}(f) \leq \frac{2(|\mathcal{S}|-flow(\mathcal{S}))}{|\mathcal{S}|} + |\mathcal{Y}|c\Delta(\mathcal{S})$.

□

Then the remaining path toward completing the proof of Theorem 3 is mostly the same as that in [23]. As our focus is the generalization risk of different subsets of training nodes rather than the performance fairness among different subgroups of test nodes, we no longer need to index different subgroups yet use $\mathcal{S}_k, k \leq n$

to index the subsets produced by our algorithm, where $|S_k| = k$. It is worth noticing that $k$ is not allowed to be arbitrarily close to zero. In practice, data pruning is promising when $n$ is extremely large, namely, there is a lot of redundancy in the data. However, it certainly leads to performance corruption if the remaining training sample is too small. Moreover, our theoretical result has a requirement that $|S|$ should be large enough, where the specific sense will be detailed later.

Then, based on the above assumptions and lemma, our counterpart of Lemma 6 in [23] should be stated as follows:

**Lemma 6** ([23]). *Under the Assumption 5.1 and 5.2, for any of our considered subset $S_k$, any $0 < \lambda \leq k^{2\alpha}$ and any $\gamma \geq 0$, Prior distribution $P$ on $\mathcal{H}$ is defined as in Assumption 5.2. Then we have:*

$$\ln \mathbb{E}_{h \sim P}[e^{\lambda(\mathcal{L}_{\mathcal{V}^{(ts)}}^{\gamma/4}(h) - \mathcal{L}_{S_k}^{\gamma/2}(h))}]$$
$$\leq \ln 3 + \lambda\left(\frac{2(k - flow(S_k))}{k} + |\mathcal{Y}|c\Delta(S_k)\right).$$

The proof is the same as that in [23] except for the difference in assumed value that will be used in the derivation. Hence, we omit a copy of it here.

Again for the same reason, an intermediate result of the Theorem 1 in [26] needs to be changed according to our cases as follows:

**Lemma 7** ([26]). *Let $\tilde{h}$ be any classifier in $\mathcal{H}$ with parameters $\{\tilde{W}_l\}_{l=1}^L$. Define $\tilde{\beta} = (\prod_{l=1}^L \|\tilde{W}_l\|_2)^{\frac{1}{L}}$. Let $\{U_l\}_{l=1}^L$ be the random perturbation to be applied to $\{\tilde{W}_l\}_{l=1}^L$, and $\tilde{U}_l \sim \mathcal{N}(0, \sigma^2 I)$. Again define $B_k$ as in Assumption 5.3. If for any of our considered subset $S_k$:*

$$\sigma \leq \frac{\gamma}{84LB_k\beta^{L-1}\sqrt{b \ln 4bL}},$$

*and $\beta$ is any constant satisfying $|\tilde{\beta} - \beta| \leq \frac{\tilde{\beta}}{L}$, then regarding the randomness of $\{U_l\}_{l=1}^L$:*

$$P\left(\max_{i \in \mathcal{V}^{(ts)} \cup S_k} \|\tilde{h}_i(H; \{\tilde{W}_l\}_{l=1}^L) - \tilde{h}_i(H; \{\tilde{W}_l + U_l\}_{l=1}^L)\|_\infty < \frac{\gamma}{8}\right) > \frac{1}{2},$$

Similarly, the Theorem 5 in [23] needs to be changed according to our cases as follows:

**Theorem 8** ([23]). *Any $\tilde{h} \in \mathcal{H}$, for any of our considered subset $S_k$, for any $\lambda > 0$ and $\gamma \geq 0$, for any "prior" distribution $P$ on $\mathcal{H}$ that is independent of $S_k$, with probability $1 - \delta$ over the sample of $y_i, i \in S_k$, for any probability distribution $Q$ on $\mathcal{H}$ s.t., $P_{h \sim Q}\left(\max_{i \in \mathcal{V}^{(ts)} \cup S_k} \|h_i(H) - \tilde{h}_i(H)\|_\infty < \frac{\gamma}{8}\right) > \frac{1}{2}$, we have*

$$\mathcal{L}_{\mathcal{V}^{(ts)}}^0(\tilde{h}) \leq \hat{\mathcal{L}}_{S_k}^\gamma(\tilde{h}) + \frac{1}{\lambda}\left(2(D_{KL}(Q\|P) + 1) + \ln\frac{1}{\delta} + \frac{\lambda^2}{4k}\right.$$
$$\left. + \ln \mathbb{E}_{h \sim P} e^{\lambda(\mathcal{L}_{\mathcal{V}^{(ts)}}^{\gamma/4}(h) - \mathcal{L}_{S_k}^{\gamma/2}(h))}\right).$$

Finally, combining all the above assumptions and results, we reach our Theorem 3, which we re-state as follows:

**Theorem 9.** *For any $\tilde{h} \in \mathcal{H}$ with parameters $\{\tilde{W}_l\}_{l=1}^L$. Under Assumptions 5.1,5.2, and 5.3, for any of our considered subset $S_k$ that ensures $k$ large enough, for any $\gamma \geq 0$, with probability at least $1 - \delta$ over the sample of $y_i, i \in S_k$, we have*

$$\mathcal{L}_{\mathcal{V}^{(ts)}}^0(\tilde{h}) \leq \hat{\mathcal{L}}_{S_k}^\gamma(\tilde{h}) + O\left(\frac{b\sum_{l=1}^L \|\tilde{W}_l\|_F^2}{k^\alpha(\gamma/8)^{2/L}}\Delta(S_k)^{2/L} + \right.$$
$$\frac{1}{k^{2\alpha}}\ln\frac{LC(2B_k)^{1/L}}{\gamma^{1/L}\delta} + \frac{1}{k^{1-2\alpha}} +$$
$$\left.\frac{(k - flow(S_k))}{k} + |\mathcal{Y}|c\Delta(S_k)\right).$$

PROOF. The logic for proving our theorem is the same as that for proving the Theorem 6 in [23]. We just elaborate on the equations with different terms in our cases, and highlight some crucial differences.

In our cases, for a specific $\beta$, to satisfy Lemma 6 and Lemma 7, we would let $\sigma$ take (at most):

$$\min\left(\frac{(\gamma/8\Delta(S_k))^{1/L}}{\sqrt{2b(\lambda k^{-\alpha} + \ln 2bL)}}, \frac{\gamma}{84LB_k\beta^{L-1}\sqrt{b \ln 4bL}}\right).$$

By considering a prior distribution $P$ sampling vectorized MLP parameters from $\mathcal{N}(0, \sigma^2 I)$ and a posterior $Q$ that adds perturbation obeying the same distribution as $P$ to $\{\tilde{W}_l\}_{l=1}^L$, then for any $\tilde{h}$ whose $\{\tilde{W}_l\}_{l=1}^L$ has a $\tilde{\beta}$ satisfying $|\tilde{\beta} - \beta| \leq \frac{\tilde{\beta}}{L}$, Lemma 7 tells us that the condition Theorem 8 requires is satisfied. Meanwhile, as Lemma 6 has been satisfied, with probability $1 - \delta$,

$$\mathcal{L}_{\mathcal{V}^{(ts)}}^0(\tilde{h}) - \hat{\mathcal{L}}_{S_k}^\gamma(\tilde{h})$$
$$\leq \frac{1}{\lambda}\left(2(D_{KL}(Q\|P) + 1) + \ln\frac{1}{\delta} + \frac{\lambda^2}{4k} + \right.$$
$$\left.\ln 3 + \lambda\left(\frac{2(k - flow(S_k))}{k} + |\mathcal{Y}|c\Delta(S_k)\right)\right)$$
$$= \frac{2}{k^{2\alpha}}D_{KL}(Q\|P) + \frac{1}{k^{2\alpha}}(\ln\frac{3}{\delta} + 2) + \frac{1}{4k^{1-2\alpha}} +$$
$$\left(\frac{2(k - flow(S_k))}{k} + |\mathcal{Y}|c\Delta(S_k)\right),$$

where we take $\lambda = k^{2\alpha}$ as Lemma 6 allows.

**Different from [23], which just needs $|\mathcal{V}^{(tr)}|$ to be large enough, we should restrict our scope to subsets that have large enough cardinality.** Specifically, the considered $k$ should be large enough to satisfy:

$$\frac{(\gamma/8\Delta(S_k))^{1/L}}{\sqrt{2b(k^\alpha + \ln 2bL)}} < \frac{\gamma}{84LB_k\beta^{L-1}\sqrt{b \ln 4bL}},$$

which is viable since $B_k$ and $\beta$ are upper bounded by Assumption 5.3.

Then, combining the fact that $D_{KL}(Q\|P) < \frac{\sum_{l=1}^L \|\tilde{W}_l\|_F^2}{2\sigma^2}$ and the considered value of $\sigma$, we have that, with probability at least $1 - \delta$,

$$\mathcal{L}^0_{\mathcal{V}^{(\text{ts})}}(\tilde{h}) - \hat{\mathcal{L}}^{\gamma}_{\mathcal{S}_k}(\tilde{h})$$

$$\leq \frac{2b(k^{\alpha} + \ln 2bL)\sum_{l=1}^{L}\|\tilde{W}_l\|_F^2}{k^{2\alpha}(\gamma/8\Delta(\mathcal{S}_k))^{2/L}} + \frac{1}{k^{2\alpha}}(\ln\frac{3}{\delta} + 2) + \frac{1}{4k^{1-2\alpha}} +$$

$$\left(\frac{2(k - flow(\mathcal{S}_k))}{k} + |\mathcal{Y}|c\Delta(\mathcal{S}_k)\right)$$

$$\leq O\left(\frac{b\sum_{l=1}^{L}\|\tilde{W}_l\|_F^2}{k^{\alpha}(\gamma/8)^{2/L}}\Delta(\mathcal{S}_k)^{2/L} + \frac{1}{k^{2\alpha}}\ln\frac{1}{\delta} + \frac{1}{k^{1-2\alpha}} + \right.$$

$$\left.\frac{(k - flow(\mathcal{S}_k))}{k} + |\mathcal{Y}|c\Delta(\mathcal{S}_k)\right).$$

The remaining part is the same as [23], which, in our case, needs to replace $\delta$ by $\frac{\gamma^{1/L}\delta}{LC(2B_k)^{1/L}}$. Thus, the final result becomes:

$$\mathcal{L}^0_{\mathcal{V}^{(\text{ts})}}(\tilde{h}) - \hat{\mathcal{L}}^{\gamma}_{\mathcal{S}_k}(\tilde{h})$$

$$\leq O\left(\frac{b\sum_{l=1}^{L}\|\tilde{W}_l\|_F^2}{k^{\alpha}(\gamma/8)^{2/L}}\Delta(\mathcal{S}_k)^{2/L} + \frac{1}{k^{2\alpha}}\ln\frac{LC(2B_k)^{1/L}}{\gamma^{1/L}\delta} + \frac{1}{k^{1-2\alpha}} + \right.$$

$$\left.\frac{(k - flow(\mathcal{S}_k))}{k} + |\mathcal{Y}|c\Delta(\mathcal{S}_k)\right).$$

$\square$

## C  MORE EXPERIMENTAL RESULTS

Considering that different representations may have an impact on the performance of Algorithm 1, we conduct comparative analysis by executing Algorithm 1 on ogbn-products, using our representation and the representation pre-trained by GVAE [17], respectively, and comparing their resulting ranking list of training nodes. For each ranking list, how model performance scales is shown in Figure 8. Clearly, the pre-trained representation is surpassed by our simple yet effective random walk-based representation $H$. Out of our expectation, performance with the pre-trained representation is even worse than random pruning in some intervals. GVAE [17] is generally used to restore the edges between nodes, therefore, the information contained in the representation pre-trained by GVAE [17] may be not suitable for node classification tasks.

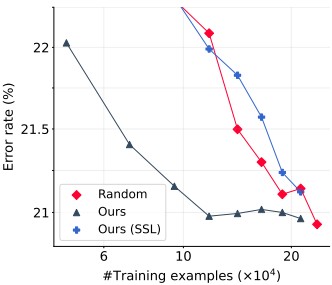

Figure 8: Comparison of different representations $T$.

