# OpenReview forum: "Exploring Neural Scaling Law and Data Pruning Methods For Node Classification on Large-scale Graphs"
_ACM.org/TheWebConf/2024/Conference — TheWebConf24 Oral_

### Official Review · Reviewer_22N1 · 2023-11-14

**Novelty:** 4
**Technical Quality:** 6

**Review:**

This paper explores data pruning methods for node classification tasks using graph neural networks. Following the observation in image and language models, the paper investigates the presence of neural scaling law in GNNs. Based on their affirmative finding of neural scaling law in GNNs, the authors developed a pruning technique to remove training nodes without sacrificing the test accuracy. Overall, this is a well written paper with clear objectives and conclusions.

Strengths:
1. The paper explored a clearly articulated problem for graphs. Even though this problem has been studied extensively in image and language models, its applications in graphs are not explored adequately.

2. The benchmarking section of the paper (Section 4) is nicely done and provides interesting insights on data pruning for node classification tasks.



Weaknesses:

Some of experimental settings are not clearly mentioned. Did you use semi-supervised settings in the experiment shown in Fig. 1? For example, when 10% training nodes are used in the training, do you remove all other training nodes and their adjacent edges from the graph? The validity of experiments depends on this information.

Figure 1 needs to capture uncertainty coming from random node selection. For example, if we select a different 10% nodes at random, do we still see the same error rate? I think each data point needs at least ten repetitions to get reasonable statistics.


One needs to use more datasets to ensure that conclusions are generally applicable for most real-world graphs including homophilic and heterophilic graphs. The paper only explores one type of graph from the OBGN family.

One of the main reasons for pruning is to train a GNN faster. The paper needs to show training time with and without pruning to justify the need for pruning.



Other comments:

Section 3 last paragraph: “Slow scaling implies that a significant fraction of training examples and computations are allocated for marginal performance gains.” I am not sure if we can make such general conclusion without adequate evidence. Does it depend on homophily or heterophilic nature of the graph? The paper only explores one type of graph from the OBGN family. Hence, the conclusions are not general enough.

**Questions:**

1. Did you use semi-supervised settings in the experiment shown in Fig. 1? For example, when 10% training nodes are used in the training, do you remove all other training nodes and their adjacent edges from the graph?

2. Did your pruning algorithm reduce the training time of GNNs?

3. Is there any relationship between pruning and explanability of GNNs?

**Reviewer Confidence:**

3: The reviewer is confident but not certain that the evaluation is correct

**Scope:**

4: The work is relevant to the Web and to the track, and is of broad interest to the community

---

### Official Review · Reviewer_hiLJ · 2023-11-17

**Novelty:** 5
**Technical Quality:** 6

**Review:**

This paper proposes a novel method for pruning nodes from large graph datasets. They design a greedy min-max algorithm to select a subset from the training nodes that are most similar to the test set.

Strength:
1. The paper is easy to follow, and the discussion on related works and the analysis of preliminary tests are sufficient.
2. The proposed algorithm achieves excellent results, validating the ability to save computational time on large graph datasets.

Weakness:
1. The connection between the proposed method and the previous discussion is unclear to me. Please see the question part for details.
2. Limitations should be discussed, for example, this method can only work under the transductive setting due to the reliance on test nodes during training.

**Questions:**

1. The authors mentioned keeping hard samples in the training set contributes to the successful pruning strategy, and the proposed method aims to choose the training samples that are most similar to the test nodes. Could you explain the connection between these two concepts?

**Reviewer Confidence:**

3: The reviewer is confident but not certain that the evaluation is correct

**Scope:**

4: The work is relevant to the Web and to the track, and is of broad interest to the community

---

### Official Review · Reviewer_jRTR · 2023-11-22

**Novelty:** 5
**Technical Quality:** 5

**Review:**

This paper studies the neural scaling property on web-scale graphs. Specifically, the authors explore SOTA data pruning methods to gain insights for effective training node selection. Subsequently, a new data pruning method is proposed.


Strength:

1. Necessary references are cited in the introduction section to justify the motivation of this work, as well as the experimental results.
2. The influence of the selected subset training set on the loss function is rigorously analyzed and proved.
3. Substantial experiments are conducted to support the claim.


Weakness:
1. This paper is relatively notation-heavy, thus not easy to follow. A notation table is suggested. For example, $\gamma$ is defined in Section 2 and then reused in Section 5 without explicit clarification.
2. Explanations of rationale or intuition of Theorems or lemmas are required for better understanding.
3. Theorem 3 relies on three assumptions, which raises concerns about its practical applicability. A clarification or justification is suggested.

Minor comments:
1. As far as I know, the training set in Papers100M accounts for around 1% or so. How is the result of Figure(1) b derived?

**Questions:**

Please refer to the weakness and minor comments.

**Reviewer Confidence:**

2: The reviewer is willing to defend the evaluation, but it is likely that the reviewer did not understand parts of the paper

**Scope:**

3: The work is somewhat relevant to the Web and to the track, and is of narrow interest to a sub-community

---

### Official Review · Reviewer_tLLU · 2023-11-24

**Novelty:** 5
**Technical Quality:** 5

**Review:**

In this paper, the authors first investigate the effectiveness of neural scaling law and data pruning methods for the node classification task on graphs. Then they develop a pruning method whose key idea is to retain those that are more similar to testing nodes to reduce the training cost.

Pros:

1.	This paper studies an interesting problem of selecting a subset of nodes from the training set to reduce the training cost.

2.	This paper further develops a new pruning strategy that prioritizes those that are more similar to testing nodes.

3.	The paper provides detailed theoretical analyses of the proposed method.

Cons:

1.	The effectiveness of the proposed method appears to exhibit a slight margin when compared to existing methods.

2.	Lacking necessary experiments to demonstrate the efficiency of the proposed method.

3.	Some formulas are not numbered.

**Questions:**

1.	I am concerned about the proposed method's computational cost on large-scale graphs. Although Line 661 shows the complexity of the proposed method, I suggest the authors report the running cost of the proposed method on benchmark datasets.

2.	I think the purpose of the proposed method is to reduce the training cost of GNN methods. Based on this opinion, I suggest the authors conduct experiments to compare the training cost using or not using the data pruning method.

3.	The results on the ogbn-products shown in Fig. 4 seem to have several mistakes.

**Reviewer Confidence:**

2: The reviewer is willing to defend the evaluation, but it is likely that the reviewer did not understand parts of the paper

**Scope:**

4: The work is relevant to the Web and to the track, and is of broad interest to the community

---

### Official Review · Reviewer_iAyg · 2023-12-01

**Novelty:** 5
**Technical Quality:** 5

**Review:**

Summary
The authors investigate an important and interesting problem, which is to explore the neural scaling law in large graphs. They conduct benchmark tests on several state-of-the-art data pruning methods for node classification tasks, thereby validating the potential of using data redundancy to improve the original undesirable power law and obtaining representative principles for selecting effective training node subsets. Additionally, this work proposes a novel data pruning method that achieves good results on all three datasets.

Strengths
1. The paper is generally well-written and easy to follow.
2. The investigated research question, neural scaling law and data pruning methods for
node classification on large-scale graphs, is interesting and useful. The proposed method is easy to understand and makes sense.
3. In addition to empirical observations, the authors also provide some theoretical evidence for validation.

Weaknesses
1. In Figure 2, why were experiments conducted using GraphSAGE, SGC, and GAT on three different datasets? Would applying GraphSAGE to these datasets yield the same results?
2. If the authors’ proposed viewpoint applies more broadly, beyond just node classification tasks, it will have a broader impact on the graph learning research community.

**Questions:**

N/A

**Reviewer Confidence:**

3: The reviewer is confident but not certain that the evaluation is correct

**Scope:**

4: The work is relevant to the Web and to the track, and is of broad interest to the community

---

### Decision · Program_Chairs · 2024-01-22

**Decision:**

Accept (Oral)

**Comment:**

This is an interesting paper that investigates scaling laws, characterizing the training error versus the test sample set size, for node classification in large graphs. The paper provides both convincing experimental and some basic theoretical results in support of the findings. It also uses the results to propose effective node pruning strategies. All reviewer's were in agreement that this is relevant and technically solid work, and I recommend acceptance. I also appreciate the very thorough and detailed response by the authors.